# Prevalence of Upper Gastrointestinal Symptoms and Gastric Dysrhythmias in Diabetic and Non-Diabetic Indian Populations: A Real-World Retrospective Analysis from Electrogastrography Data

**DOI:** 10.3390/diagnostics15070895

**Published:** 2025-04-01

**Authors:** Sanjay Bandyopadhyay, Ajit Kolatkar

**Affiliations:** 1Kolkata Gastro Care, Jessore Road, Dumdum, Kolkata 700028, West Bengal, India; 2GastroLab India Pvt. Ltd., 202, Speciality Business Centre, Balewadi Rd, Pune 411045, Maharashtra, India; dr.ajit@gastrolab.com

**Keywords:** gastroparesis, dyspepsia, electrogastrography, gastric myoelectric activity, diabetes, upper GI disorders

## Abstract

**Background:** Upper gastrointestinal (GI) motility disorders, such as gastroparesis and functional dyspepsia (FD), contribute significantly to morbidity, especially in populations at risk for type 2 diabetes. However, the prevalence and clinical manifestations of these disorders in India, and associated gastric dysrhythmias, are not well studied within this population. **Methods:** This retrospective, cross-sectional study analyzed 3689 patients who underwent electrogastrography with water load satiety test (EGGWLST) testing across multiple motility clinics in India. The prevalence of gastroparesis and FD-like symptoms, symptom severity, and their association with diabetes and other comorbidities were evaluated. Symptom severity was assessed using the Gastroparesis Cardinal Symptom Index (GCSI). EGGWLST findings were documented, including the gastric myoelectric activity threshold (GMAT) scores. **Results:** The study population had a mean age of 43.18 years. GCSI scores indicated that patients had symptoms that were mild (55%), moderate (33%), and severe (8%). Compared with the non-diabetic population, diabetic subjects had significantly higher rates of early satiety (56% vs. 45%, *p* < 0.0001), bloating (73% vs. 67%, *p* = 0.005), and reflux (28% vs. 24%, *p* = 0.029). WLST data analysis revealed that significantly more diabetic subjects ingested <350 mL (16% vs. 12%, *p* = 0.000016). EGG analysis revealed gastric dysthymias in one-third (65%) of patients. Significantly more diabetic subjects (22% vs. 18% *p* = 0.015) had a GMAT score >0.59. **Conclusions:** Upper GI motility disorders are prevalent in India, particularly among diabetic patients. EGG is a valuable tool for characterizing these disorders, and may help in personalizing therapeutic approaches. Further research is required to optimize treatment strategies.

## 1. Introduction

Upper gastrointestinal (GI) motility disorders including gastroparesis (GP) and functional dyspepsia (FD) are significant contributors to morbidity worldwide, particularly among individuals with diabetes [1]. Patients with upper GI disorders commonly present with symptoms such as bloating, early satiety, and nausea, which can impact their quality of life and lead to substantial healthcare costs [2]. GP and FD have traditionally been distinguished by the presence or absence of delayed gastric emptying. Gastroparesis-like symptoms have a global prevalence of 0.9%, with a higher rate of 1.3% among diabetic individuals [3]. In the Asian population, the prevalence of GP may be significantly underestimated. Asian individuals are at an increased risk of developing type 2 diabetes, and research by Oshima et al. indicates that 13.8–29% of type 2 diabetic patients with upper GI symptoms may experience delayed gastric emptying [4]. Chronic upper GI symptoms and FD are prevalent among Asians, with a survey of 490 Asian doctors revealing that 47.2% suspect that 25–45% of patients diagnosed with FD may suffer from GP. Considering that FD affects 8–23% of the Asian population, and that 23–35% of these patients exhibit delayed gastric emptying, the estimated prevalence of gastroparesis in this demographic ranges from 1.84% to 8.05% [5]. Despite its local and global relevance, the prevalence and clinical characteristics of these conditions remain unexplored in specific populations, including those with metabolic comorbidities.

GP involves delayed gastric emptying without mechanical obstruction as a result of neuromuscular dysfunction. In contrast, FD, or non-ulcer dyspepsia, is classified as a disorder of the gut–brain interaction. Accumulating evidence suggests significant overlap in the clinical presentation, symptoms, and treatment outcomes of these conditions, and the notion that GP and FD belong to the same clinicopathological spectrum has been gaining prominence. However, this classification remains a subject of ongoing debate. For instance, this perspective is supported by findings from a prospective study conducted by Pasricha PJ et al. that included 944 patients over a 12-month period at a tertiary care center. In this study, patients were classified as having gastroparesis if gastric emptying was delayed. Otherwise, they were labeled as having FD based on the Rome III criteria. After a year of follow-up, patients with FD and GP were indistinguishable based on clinical and pathologic features, as well as assessments of gastric emptying. These findings reinforce the notion that GP and FD are interchangeable at tertiary care centers and underscore the importance of considering them within the same clinicopathological spectrum [6].

Gastric accommodation is a vital process that enables the fundus and proximal gastric body to relax properly, allowing the stomach to hold and process ingested food. This function is triggered by food intake and regulated by a nitric oxide-mediated vagal reflex [7]. Additionally, antral distension triggers an antro-fundic reflex, further facilitating gastric accommodation. Studies have shown that impaired gastric accommodation occurs in 40% of patients with FD and in 43% of those with idiopathic GP [8]. Damage to the vagus nerve or autonomic dysfunction may lead to impaired gastric accommodation, causing symptoms such as early satiety and postprandial fullness in these patients. Antral hypomotility is a common gastric motor dysfunction in FD and GP, as demonstrated by antroduodenal manometry. Its cause may be primary or related to antral distension and impaired gastric accommodation. The interstitial cells of Cajal (ICCs), essential for regulating gastric contractions, are reduced in severe diabetic and idiopathic gastroparesis, suggesting a potential cellular basis for this hypomotility [9]. The diagnosis and management of these motility disorders are complicated by the nonspecific nature of symptoms and overlapping presentation with other GI conditions [10]. Electrogastrography (EGG), a non-invasive tool that measures gastric myoelectric activity (GMA), offers the potential to distinguish between different motility patterns, such as normal GMA; dysrhythmic GMA like tachygastria, bradygastria, and antropyloroduodenal dysfunction (APD); or accommodation dysfunction [11]. EGG is recommended as a diagnostic tool for evaluating patients with unexplained nausea, vomiting, and other dyspeptic symptoms to better understand the underlying mechanisms of these symptoms [12]. However, its utility in routine clinical practice, especially in the diabetic population, remains understudied.

EGG has revealed altered GMA in approximately two-thirds of patients with FD and a majority of those with GP. It identifies distinct gastric myoelectrical patterns in gastroparesis, whether due to mechanical obstruction or idiopathic causes. Abnormal EGG patterns, such as tachygastria, bradygastria, and persistent activity at 3 cpm, correlate with delayed gastric emptying and suggest neuromuscular dysfunction or gastric outflow resistance, especially in idiopathic cases [13]. EGG measures GMA in response to the water load satiety test (WLST). In healthy individuals, the WLST typically evokes a normal 3 cpm GMA response. However, patients with GP and FD often display hyponormal 3 cpm GMA and various gastric dysrhythmias, which are frequently associated with the reduction in ICCs in the gastric wall [14]. Classical and high-resolution EGG studies have demonstrated a significant link between gastric dysrhythmia and nausea, a key symptom of GP. Collectively, these findings underscore the critical role of gastric dysrhythmias in the pathophysiology of FD and GP [9]. The absence of ICCs has been linked to increased abnormalities in gastric slow waves, more severe symptoms, and a reduced response to gastric electrical stimulation (GES). EGG may serve as a clinical marker for the depletion of ICCs and could potentially predict treatment response to GES [15].

The GMAT score derived from the 3 cpm GMA responses to WLST may play a critical role in identifying a subset of patients with underlying APD dysfunction or gastric outflow dysfunction. Studies, including Noar et al., have shown that patients identified using a GMAT score may respond positively to pyloric-directed therapy, like balloon dilation, with a clinical success rate of up to 93% [14]. This provides preliminary evidence suggesting that the GMAT score is a valuable prognostic and predictive tool for tailoring therapeutic interventions for patients with GP and possibly FD.

India, with its high burden of type 2 diabetes mellitus, presents a unique opportunity to study the intersection of diabetes and GI motility disorders. Given the distinct regional dietary habits and metabolic risk profiles in this population, understanding the prevalence of GMA dysrhythmias and their influence on upper GI symptoms may help tailor more region-specific treatment strategies. This study aims to retrospectively analyze the prevalence of upper GI motility disorders in the diabetic and non-diabetic population in India using EGG data to evaluate GMA subtypes and characteristics, and their relationship to specific symptoms. By investigating the association between GMA abnormalities and symptom severity, we aim to provide insights that could lead to more personalized diagnostic and therapeutic approaches in clinical practice.

## 2. Materials and Methods

This retrospective cross-sectional, multicenter study analyzed data from subjects who underwent EGG tests at motility clinics in various cities across India. The study aimed to determine the prevalence of GP-like and FD-like disorders among Indian patients. It also explored the age distribution, gender preponderance, EGG characteristics, and associations of patient-reported upper GI motility disorders and related symptoms with comorbidities such as diabetes. The study adhered to the principles outlined in the Declaration of Helsinki, the International Council for Harmonization–Good Clinical Practice (ICH–GCP) guidelines, the Indian Council of Medical Research, and the Indian GCP guidelines, in accordance with the approved protocol. The data analysis process commenced only after receiving approval from the independent ethics committee (IEC NO. CIEC081024). As data for this study was retrospectively collected, informed consent was not required. Patient confidentiality was maintained throughout the data entry and analysis process.

### 2.1. Electrogastrography (EGG)

EGG with water load satiety test (EGGWLST) is a standardized, non-invasive diagnostic modality used to measure the GMA [11]. It was recorded with a USFDA-approved, validated electrogastrography device made by 3CPM EGG, Sparks, MD, USA. Subjects were required to fast for 6–8 h, after which 10 min baseline readings were taken. They then ingested water until completely full over a five-minute period, after which WLST EGG readings were recorded for the next 30 min. Water ingestion of <350 mL was considered abnormal. GMA percentage distribution of power was recorded in four frequency ranges: normal, 2.5–3.75 cpm; bradygastria, 1–2.5 cpm; tachygastria, 3.75–10 cpm; and duodenal respiration, 10–15 cpm. GMA percent frequency distribution of power of patients across all frequencies and time points was compared with control values to determine each patient’s GMA response to WLST.

### 2.2. Data Collection and Analysis

Qualitative (categorical) and quantitative (continuous) variables were analyzed using descriptive statistics. Quantitative variables were evaluated using *t*-tests or ANOVA, while qualitative variables were assessed using Chi-square tests to determine relationships between variables in the study population. The corresponding *p*-values are reported. Data analysis was performed using GraphPad statistical software version 10.4.1.

## 3. Results

### 3.1. Description of Demographic and Symptom Characteristics in Overall Population and Between Diabetic Versus Non-Diabetic Populations

The study included 3689 patients with a mean age of 43.18 years. Diabetic patients (*n* = 714) were significantly older than non-diabetics (56 vs. 40 years, *p* < 0.0001). Gender distribution, detailed in Table 1, showed a slightly higher proportion of males (55%) overall. The most commonly reported symptoms (Figure 1) were bloating (68%), early satiety (46%), and postprandial fullness (42%). Diabetic patients had a higher prevalence of early satiety (56% vs. 45%, *p* < 0.0001), bloating (73% vs. 67%, *p* = 0.0015), reflux (28% vs. 24%, *p* = 0.029), and constipation (35% vs. 30%, *p* = 0.006), while non-diabetics reported more epigastric pain (20% vs. 13%, *p* = 0.0003).

When evaluating the severity of GI symptoms using the GCSI score, most participants reported symptoms that were mild (55%), followed by moderate (33%), and severe (8%). The distribution of symptom severity was comparable between the non-diabetic and diabetic groups (Table 1).

### 3.2. Regional Analysis of Demographic and Symptom Characteristics

The study analyzed the demographic characteristics and GI symptoms across four regions of India, namely, North, South, West, and East, revealing significant differences among these regions (Table 2). Age distribution differed significantly, with median ages ranging from 42 years in the North to 44 years in the South (*p* = 0.0215). The gender distribution was similar across regions, with a slight male predominance. GI symptoms varied significantly across regions (Table 2). Early satiety was most prevalent in the East (66%) and least prevalent in the West (25%) (*p* < 0.0001). Postprandial fullness was more prevalent in the West (56%) and was lowest in the South (31%) (*p* < 0.0001). Bloating was highest in the South (76%) and lowest in the West (57%) (*p* < 0.0001). Abdominal pain was most common in the North (36%) (*p* = 0.0187), while epigastric pain and burning were highest in the West (29% and 22%, respectively) (*p* < 0.0001). Anorexia and weight loss were most frequent in the East (16% and 29%, respectively) (*p* < 0.0001). Loss of appetite was highest in the North (37%) and lowest in the South (19%) (*p* < 0.0001). Constipation was more common in the West (44%) (*p* < 0.0001), while reflux was most common in the South (45%) and East (44%) (*p* < 0.0001). Nausea and vomiting varied significantly: nausea was highest in the East (43%) and vomiting was highest in the West (29%) (*p* = 0.002). These findings highlight notable regional differences in GI symptoms (Figure 2).

### 3.3. Water Ingestion and Gastric Myoelectric Activity Characteristics

Table 3 details the water ingestion volumes, with the average water ingestion across the study population reported as 533.51 mL. Though there were differences found in average water ingestion, their values did not reach statistical significance. Subgroup analysis revealed that 87% of participants ingested more than 350 mL of water, with a higher proportion of non-diabetics (87%) ingesting this amount compared with diabetics (83%) (*p* = 0.00027). Diabetic participants were more likely to consume less than 350 mL (17% vs. 11%) (*p* = 0.000016).

In the overall population, 20% had a GMAT score >0.59, with a significantly higher proportion of diabetics (22%) having this score compared with non-diabetics (18%) (*p* = 0.015). In total, 25% had a GMAT score <0.59, with no statistical differences found in diabetics (24%) compared with non-diabetics (25%). (Table 3).

A dysrhythmic GMA response was observed in over 65% of subjects, with tachygastria observed in 37%, bradygastria in 22%, and mixed dysrhythmias in 6%. The dysrhythmic GMA responses were statistically similar in diabetics versus non-diabetics as detailed in Table 3. Hyponormal 3 cpm GMA was observed in 55%, with no significant group differences (*p* = 0.97). In normal 3 cpm GMA responses, 13% had normal GMA (*p* = 0.59), 5% had hypernormal GMA (*p* = 0.71), and 12% had normal GMA with dysrhythmia (*p* = 0.88), with no significant differences. APD dysfunction was seen in 20%, with the condition more common in diabetics (22%) than non-diabetics (18%) (*p* = 0.01). The dysfunction of ICCs affected 55%, with no significant differences (*p* = 0.97).

### 3.4. Assessment of Upper GI Symptom Characteristics, EG-Based GMA Subtypes and Characteristics in Diabetic Versus Non-Diabetic Groups

Table 4 details the symptom frequency distribution and its association to GMA responses to the WLST, compared between diabetic and non-diabetic groups. Diabetic patients with early satiety were significantly more likely to have tachygastria (51% vs. 42%, *p* = 0.0059), bradygastria (60% vs. 47%, *p* = 0.008), hyponormal 3 cpm GMA (51% vs. 45%, *p* = 0.027), and normal 3 cpm GMA (57% vs. 45%, *p* = 0.044) than non-diabetic patients. No significant differences were found in mixed dysrhythmia (33% vs. 43%, *p* = 0.29), hypernormal 3 cpm GMA (58% vs. 52%, *p* = 0.57), or normal 3 cpm GMA with gastric dysrhythmia (46% vs. 41%, *p* = 0.46). Bradygastria (78% vs. 69% *p* = 0.0.49) was found to be significantly associated with bloating in diabetics.

Postprandial fullness in diabetics was less likely to be associated with normal 3 cpm GMA with gastric dysrhythmias compared with non-diabetics (27% vs. 39%, *p* = 0.044). However, there was no association observed in hyponormal 3 cpm GMA (42% vs. 41% *p* = 0.647) or related predominant post-WLST dysrhythmic responses between the diabetic and non-diabetic populations (Table 4).

Abdominal pain was consistent across groups with no significant differences. Diabetic patients with normal 3 cpm GMA had significantly less epigastric pain (9% vs. 21%, *p* = 0.0097). No significant differences were found in other GMA patterns. Diabetic patients with mixed dysrhythmia were more likely to have anorexia (21% vs. 7%, *p* = 0.0076), while weight loss was more common in diabetic patients with tachygastria (13% vs. 19%, *p* = 0.0174). No significant differences were noted in other post-WLST GMA responses. Reflux and nausea/vomiting rates did not differ significantly between groups across all GMA patterns (Table 4). No other significant differences were seen in other GMA patterns.

## 4. Discussion

Upper GI symptoms are prevalent in the general population and contribute significantly to healthcare costs and lost productivity [16]. The study findings provide valuable insights into the GI symptoms and motility patterns within a large cohort of patients, including both diabetic and non-diabetic populations. In the overall population, bloating (68%), early satiety (46%), and postprandial fullness (42%) were the most reported symptoms, indicating a significant burden of GI discomfort among the study participants. Bloating is common throughout the world. Nearly 18% of the general population experiences bloating at least once per week [17]. In Asia, bloating is often reported as both an upper and lower GI symptom, with a prevalence of 26.9% [18]. A cross-sectional study of 300 participants with GI issues attending tertiary care in India found that 198 (66%) had signs of bloating and 236 (78%) had signs of postprandial fullness and early satiety [19].

Various studies have consistently demonstrated that individuals with diabetes experience a higher prevalence of both upper and lower GI symptoms compared with healthy controls [20,21,22]. The findings of our study align with this trend, showing that GI symptoms are more prevalent among diabetic patients than non-diabetic individuals. In contrast, a study using the Bowel Disease Questionnaire (BDQ) found no significant difference in the frequency of GI symptoms between those with and without diabetes [23]. Sang et al. observed that upper GI symptoms were more frequent in diabetic patients compared with non-diabetic counterparts [24]. Additionally, studies have reported a significantly higher incidence of constipation, diarrhea, alternating bowel habits, abdominal pain, eructation, and flatulence in diabetic patients compared with control groups [16,25,26]. A study conducted on an East Indian population found that early satiety was more common among diabetic females than males. Our findings corroborate this, with female patients showing a higher prevalence of early satiety (56%) compared with male patients (49%) [27]. One study indicated that the prevalence of gastroesophageal reflux symptoms in diabetic populations could be as high as 41% [28], whereas our study found a lower prevalence of 28%. We hypothesize that this lower prevalence in India may be attributable to a lower consumption of junk, processed, and reflexogenic foods compared with Western populations.

The significant regional differences in GI symptoms across North, South, West, and East India highlight varying prevalence rates of conditions such as early satiety, postprandial fullness, bloating, reflux, abdominal pain, anorexia, weight loss, constipation, and nausea. These disparities suggest that genetics, ethnicity, cultural, geographic factors, and regional food variations, including sleep hygiene, may influence symptom patterns [29,30,31]. The ROME IV Global Epidemiology Study reported a much lower prevalence of FD at 0.7% in India’s general population, based on symptom questionnaires in the community [32]. In contrast to our study, using electrogastrography assessments in symptomatic, clinic-based cohorts identified markedly higher prevalence rates of FD-related symptoms like early satiety (46%) and bloating (68%). These differences are likely due to variations in general community versus symptomatic clinical study populations. This emphasizes the need for region-specific management strategies to address the unique symptom profiles in each area.

This study explores water ingestion patterns and GI motor activity in individuals with and without diabetes. A water load test was used to assess satiety through gastric distension without triggering hormonal responses from a caloric meal [33]. In this study, 18% of diabetic participants consumed less than 350 mL of water, compared to 13% of non-diabetic participants (*p* = 0.044). Vagus nerve neuropathy in diabetes affects gastric motility and accommodation by impairing gastric relaxation [34]. Reduced nitric oxide (NO) synthesis, linked to decreased neuronal nitric oxide synthase (nNOS) expression, disrupts normal gastric accommodation and contributes to delayed gastric emptying and dysmotility, especially in GP [35]. Gastric dysrhythmias, observed in GP and FD, are associated with reduced ICCs and a decrease in normal gastric motility at 3 cpm. The loss of ICCs impairs the gastric pacemaker function, leading to dysrhythmias [36].

Gastric dysrhythmias are linked to various clinical disorders, many of which can lead to nausea and vomiting. In diabetic gastroparesis, up to 70% of patients may experience tachygastria and bradygastria [37]. Our study’s analysis of GMA subtypes revealed that tachygastria was the most prevalent dysrhythmia, affecting more than half of the participants. Among the diabetic population, the prevalence rates of tachygastria, bradygastria, and mixed dysrhythmias were 36%, 19%, and 5%, respectively.

Another important observation of our study is that early satiety is notably more prevalent in diabetic patients across various GMA subtypes, with statistically significant differences compared to their non-diabetic counterparts. This could be attributed to a greater degree of gastric neuromuscular dysfunction in diabetic patients, which significantly impacts patients’ quality of life [38].

Postprandial fullness showed no significant differences across most GMA patterns, except for normal 3 cpm GMA with dysrhythmia, where non-diabetic patients had higher rates. Bloating was more common in diabetic patients with bradygastria, suggesting a link to slowed gastric emptying. Hyponormal 3 cpm GMA showed a trend toward more bloating in diabetics, but it lacked statistical significance. Abdominal pain was consistent across both groups, indicating it may not be strongly influenced by GMA or diabetes. However, diabetic patients with normal GMA reported less epigastric pain, warranting further investigation. Increased anorexia in diabetic patients with mixed dysrhythmia suggests a connection between gastric dysrhythmias and appetite regulation. Weight loss was more common in diabetic patients with tachygastria, potentially indicating more severe gastroparesis. There were no significant differences in reflux or nausea/vomiting between groups across GMA patterns, suggesting that other factors may influence these symptoms. These findings highlight the complexity of GI symptoms in diabetes and emphasize the need for tailored management based on specific GMA patterns.

Patients with GP or FD face significant challenges in being appropriately stratified for the most effective therapeutic interventions, often leading to suboptimal health outcomes [38,39]. A subset of these patients remains refractory to standard treatments, necessitating advanced diagnostic approaches to guide management. The EGGWLST subtyping system we adopted has emerged as a promising modality for identifying patients who may benefit from targeted therapies. Notably, individuals with gastric outflow resistance or antropyloroduodenal dysfunction, characterized by a GMAT score >0.59, have shown substantial clinical improvement following pyloric-directed interventions [14]. Our data reveal that diabetic patients are disproportionately affected, with 22% of diabetics presenting with elevated GMAT scores, suggesting a higher prevalence of diabetes-induced pathophysiological alterations in the antrum and pylorus. These findings underscore the impact of diabetes on gastric motor function, particularly in the antropyloroduodenal region.

This critical patient subgroup, identified through EGGWLST subtyping and GMAT scoring, represents a population for whom pyloric-directed therapy may alleviate symptoms effectively and potentially yield long-term remission. The EGGWLST subtyping system, with its ability to distinguish antropyloroduodenal dysfunction and the dysfunction of ICCs from patients with a normal GMA response, with or without accommodation dysfunction, provides clinically actionable insights. Therefore, EGGWLST GMA-based subtyping may enable precise patient stratification, an approach that holds significant potential to improve therapeutic outcomes in GP and FD.

Overall, our study highlights the complexity of GI motor activity and dysfunction in diabetic individuals and emphasizes the clinical relevance of EGG in assessing the underlying pathophysiological change in patients presenting with upper GI symptoms. Further research is warranted to explore these relationships in greater detail, particularly regarding the role of diet, lifestyle, and other non-glycemic factors in the development of GI symptoms and dysrhythmias.

This study has several limitations. Its retrospective design may introduce selection bias as patients with more severe symptoms were more likely to be referred for EGG, potentially skewing results. The absence of gastric emptying studies limits the correlation between EGG findings and actual gastric motility. Although surface EGG has inherent limitations, including susceptibility to signal and motion artifacts, indirect measurement of gastric motility, and variations due to non-standardized electrode placement, these concerns were largely mitigated by employing well-validated, standardized study procedures conducted by well-trained personnel across all sites. Another limitation was that multivariate analysis was not applied, as symptom differences were assessed using subgroup stratification. Additionally, while multiple comparisons were conducted, the results should be interpreted in the context of clinical relevance rather that purely statistical significance. While the study offers valuable insights into the Indian population, its applicability to other populations may be limited by regional and dietary differences. Future research should focus on prospective studies that include both EGG and gastric emptying assessments for a more comprehensive understanding of gastric motility across populations.

## 5. Conclusions

This study reveals a clear association between diabetes and GI symptoms such as early satiety, bloating, and anorexia, which are associated with GMA dysrhythmias like bradygastria and tachygastria. EGG and WLST-based GMA subtyping may aid in personalizing treatments for diabetic patients with GI disorders. EGG has shown promise as a diagnostic tool for upper gastrointestinal disorders. While early satiety and bloating are more common in diabetic patients, there are no statistical differences in symptoms like abdominal pain between diabetic and non-diabetic individuals, underscoring the need for further research into the non-glycemic factors that may influence GI symptoms as well as personalized treatment strategies.

## Figures and Tables

**Figure 1 diagnostics-15-00895-f001:**
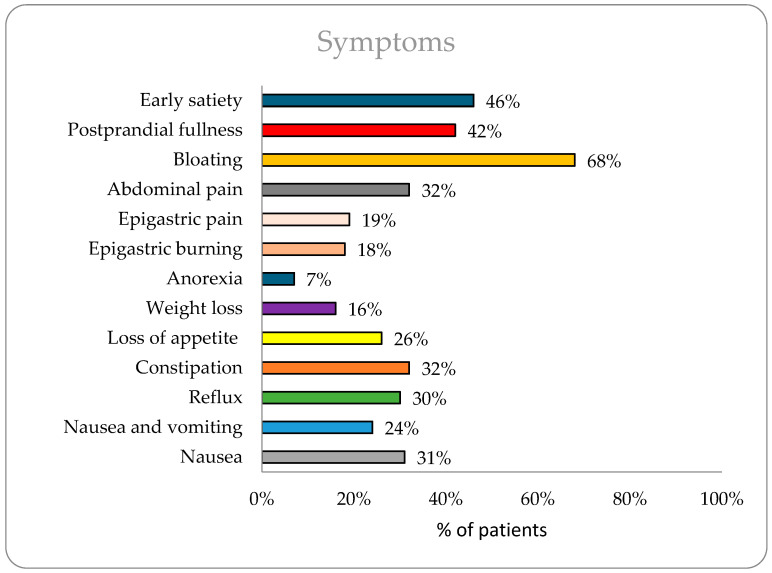
Prevalence of symptoms in overall population (*n* = 3689).

**Figure 2 diagnostics-15-00895-f002:**
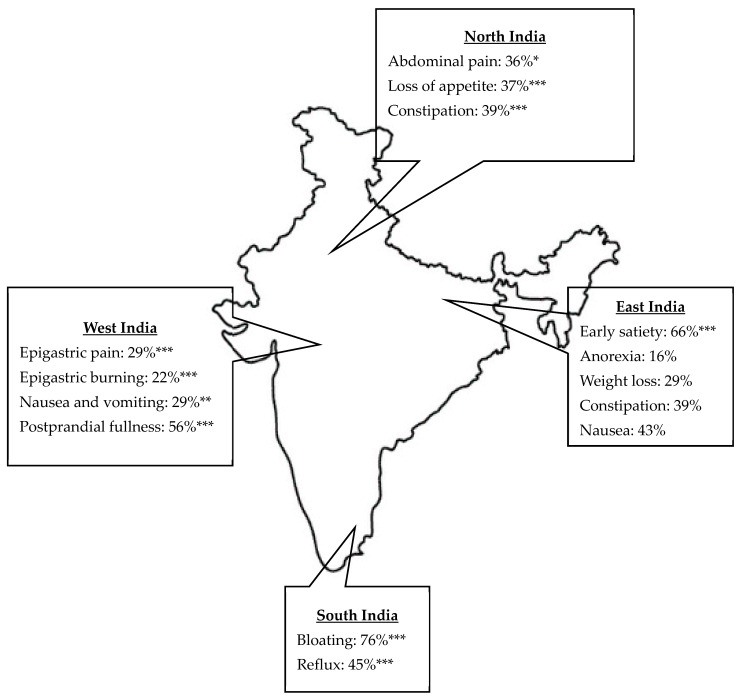
Regional variation in upper gastrointestinal symptoms in India. Only significant values are represented: * *p* < 0.05, ** *p* < 0.01, *** *p* < 0.0001 calculated using Chi-square test.

**Table 1 diagnostics-15-00895-t001:** Summary of baseline characteristics, symptoms, and symptom severity.

Variables	Overall Population (*n* = 3689)	Diabetic Population (*n* = 714)	Non-Diabetic Population (*n* = 2937)	*p*-Value (Diabetic vs. Non-Diabetic)
Age, yrs (SD)	43.18 (15.30)	56 (12)	40 (14.40)	<0.0001 ^$^
Gender				
Male, *n* (%)	2011 (55%)	365 (51%)	1627 (55%)	
Female, *n* (%)	1675 (45%)	349 (49%)	1310 (45%)	
Symptoms, *n* (%)				
Early satiety	1686 (46%)	376 (56%) ***	1310 (45%)	<0.0001
Postprandial fullness	1531 (42%)	295 (38%)	1221 (42%)	0.899
Bloating	2498 (68%)	519 (73%) **	1956 (67%)	0.0015
Abdominal pain	1191 (32%)	217 (30%)	954 (32%)	0.30
Epigastric pain	684 (19%)	100 (13%)	583 (20%) ***	0.0003
Epigastric burning	668 (18%)	124 (17%)	187 (6%)	0.83
Anorexia	244 (7%)	57 (8%)	471 (16%)	0.13
Weight loss	586 (16%)	103 (14%)	759 (26%)	0.30
Loss of appetite	963 (26%)	194 (27%)	917 (31%)	0.47
Constipation	1191 (32%)	261 (35%) **	869 (30%)	0.006
Reflux	1118 (24%)	241 (28%) *	703 (24%)	0.029
Nasuea and vomiting	879 (24%)	174 (23%)	905 (31%)	0.80
Nausea	1128 (31%)	212 (30%)	355 (13%)	0.55
GCSI score, *n* (%)				
Mild	2015 (55%)	363 (51%)	1544 (53%)	0.40
Moderate	1234 (33%)	252 (35%)	973 (33%)	0.27
Severe	299 (8%)	68 (10%)	227 (8%)	0.11

Abbreviations: GCSI—Gastroparesis Cardinal Symptom Index; SD—Standard Deviation; * *p* < 0.05, ** *p* < 0.01, *** *p* < 0.0001 calculated using Chi-square test; ^$^—*p*-value calculated using *t*-test.

**Table 2 diagnostics-15-00895-t002:** Regional demographics and prevalence of symptoms.

Variables	North (*n* = 949)	South (*n* = 1368)	West (*n* = 651)	East (*n* = 720)	*p*-Value
Age	42 (11–85)	44 (5–87)	43 (13–86)	43 (6–82)	0.0215 * ($)
Gender					
Female	420 (44%)	586 (43%)	294 (45%)	375 (52%)	
Male	527 (56%)	782 (57%)	357 (55%)	345 (48%)	
Early satiety	264 (28%)	790 (58%)	159 (24%)	473 (66%) ***	<0.0001
Postprandial fullness	435 (46%)	428 (31%)	366 (56%) ***	302 (42%)	<0.0001
Bloating	583 (61%)	1040 (76%) ***	370 (57%)	504 (70%)	<0.0001
Abdominal pain	342 (36%) *	418 (31%)	210 (33%)	216 (30%)	0.0187
Epigastric pain	227 (24%)	128 (9%)	186 (29%) ***	142 (20%)	<0.0001
Epigastric burning	160 (17%)	281 (21%)	141 (22%) ***	86 (12%)	<0.0001
Anorexia	40 (4%)	43 (3%)	46 (7%)	115 (16%) ***	<0.0001
Weight loss	85 (9%)	208 (15%)	65 (12%)	212 (29%) ***	<0.0001
Loss of appetite	352 (37%) ***	258 (19%)	185 (28%)	167 (23%)	<0.0001
Constipation	372 (39%) ***	253 (18%)	288 (44%)	278 (39%) ***	<0.0001
Reflux	125 (13%)	622 (45%) ***	53 (8%)	318 (44%)	<0.0001
Nausea and vomiting	195 (21%)	324 (24%)	187 (29%) **	170 (24%)	0.002
Nausea	294 (31%)	315 (23%)	212 (33%)	306 (43%) ***	<0.0001

* *p* < 0.05, ** *p* < 0.01, *** *p* < 0.0001 calculated using Chi-square test; $—calculated using one-way ANOVA.

**Table 3 diagnostics-15-00895-t003:** Summary of water load test and GMA response to WLST.

Variables	Overall Population (*n* = 3689)	Diabetic Population (*n* = 714)	Non-Diabetic Population (*n* = 2937)	*p*-Value (Diabetic vs. Non-Diabetic)
Amount of water ingested in mL, *n* (%)				
Average amount of water ingested	533.51 ± 216.35	543.32 ± 261.58	532.32 ± 204.16	0.22
>350 mL	3201 (87%)	590 (83%) ***	2578 (87%)	0.00027
<350 mL	472 (13%)	114 (16%) ***	342 (12%)	0.000016
Average water consumed >350 mL ± SD	579.97 ± 189.86	605.38 ± 237.26 **	579.98 ± 177.27	0.0032 ^$^
Average water consumed <350 mL ± SD	218.49 ± 77.14	217.19 ± 74.39 ***	150.67 ± 52.49	0.0001 ^$^
GMAT score				
>0.59	772 (20%)	154 (22%) *	518 (18%)	0.015
<0.59	905 (25%)	171 (21%) ***	711(20%)	0.18
Dysrhythmic GMA ^a^ response				
Tachygastria	1370 (37%)	257 (36%)	1067 (36%)	0.86
Bradygastria	795 (22%)	134 (19%)	645 (22%)	0.061
Mixed dysrhythemia	210 (6%)	43 (6%)	166 (6%)	0.70
Hyponormal 3 cpm GMA	2012 (55%)	387 (54%)	1590 (54%)	0.97
Normal 3 cpm GMA response				
Normal 3 cpm GMA	472 (13%)	88 (12%)	384 (13%)	0.59
Hypernormal 3 cpm GMA	194 (5%)	36 (5%)	158 (5%)	0.71
Normal 3 cpm with dysrhythmia ^b^	439 (12%)	85 (12%)	344 (12%)	0.88
Dysfunction, *n* (%)				
APD	772 (20%)	154 (22%) *	518 (18%)	0.01
ICCs	2012 (55%)	387 (54%)	1590 (54%)	0.97
Normal 3 cpm with and without dysrhythemia	911 (24.7%)	173 (24.22%)	728 (4.78%)	0.75

Abbreviations: GMA—gastric myoelectric activity; GMAT—gastric myoelectric activity threshold; APD—antropyloroduodenal dysfunction; ICCs—interstitial cells of Cajal; cpm—cycle per minute; * *p* < 0.05, ** *p* < 0.01, *** *p* < 0.0001 calculated using Chi-square test; ^$^—*p*-value calculated using *t*-test. ^a^ Dysrhythmic GMA and hyponormal 3 cpm occur in response to WLST. ^b^ Includes patients with tachygastria, bradygastria, and mixed dysrhythmia.

**Table 4 diagnostics-15-00895-t004:** Comparative prevalence of gastrointestinal symptoms in diabetic and non-diabetic populations across different gastric myoelectrical activity (GMA) subtypes.

	Early Satiety	PPD	Bloating	Abdominal Pain	Epigastric Pain	Epigastric Burning	Anorexia	Weight-Loss	LOA	Constipation	GERD	CNV	Nausea
Diabetes + Tachygastria (*n* = 286)	147 (51%) *	101 (35%)	209 (73%)	85 (30%)	36 (13%)	47 (16%)	22 (8%)	36 (13%)	73 (26%)	99 (35%)	90 (31%)	66 (23%)	82 (29%)
No diabetes + Tachygastria (*n* = 1067)	449 (42%)	411 (39%)	718 (67%)	332 (31%)	178 (17%)	187 (18%)	64 (6%)	199 (19%) *	264 (25%)	330 (31%)	310 (29%)	253 (24%)	342 (32%)
*p* value	0.0059	0.337	0.0728	0.72	0.1	0.72	0.33	0.0174	0.82	0.25	0.42	0.87	0.28
Diabetes + Brady-gastria (*n* = 134)	80 (60%) **	63 (47%)	104 (78%) *	41 (31%)	23 (17%)	31 (23%)	8 (6%)	24 (18%)	40 (30%)	52 (39%)	46 (34%)	26 (19%)	47 (35%)
No diabetes + Brady-gastria (*n* = 1067)	317 (47%)	292 (43%)	465 (69%)	221 (33%)	150 (22%)	119 (18%)	44 (7%)	80 (12%)	156 (23%)	215 (32%)	214 (32%)	144 (21%)	206 (31%)
*p* value	0.008	0.44	0.049	0.684	0.21	0.144	>0.99	0.066	0.098	0.13	0.54	0.72	0.31
Diabetes + mixed dys-rhythemia (*n* = 43)	14 (33%)	17 (40%)	25 (58%)	17 (40%)	10 (24%)	7 (17%)	9 (21%) **	7 (17%)	12 (29%)	21 (49%)	10 (24%)	14 (33%)	10 (24%)
No diabetes + mixed dysrhythemia (*n* = 166)	71 (43%)	63 (38%)	112 (67%)	55 (33%)	45 (27%)	22 (13%)	11 (7%)	19 (11%)	41 (25%)	54 (33%)	43 (26%)	32 (19%)	43 (26%)
*p* value	0.29	0.86	0.28	0.37	0.37	0.37	0.0076	0.43	0.69	0.073	0.84	0.061	0.85
Diabetes + Hyponormal (*n* = 391)	199 (51%) *	166 (42%)	281 (72%)	126 (32%)	62 (16%)	67(17%)	28 (7%)	56 (14%)	111 (28%)	158 (40%) **	124 (32%)	96 (25%)	131 (34%)
No diabetes + hyponormal (*n* = 1596)	713 (45%)	657 (41%)	1072 (67%)	514 (32%)	323 (20%)	289 (18%)	110 (7%)	214 (13%)	378 (24%)	510 (32%)	479 (30%)	374 (23%)	518 (32%)
*p* value	0.027	0.647	0.069	>0.99	0.06	0.71	0.82	0.622	0.057	0.0019	0.53	0.64	0.72
Diabetes + normal GMA (*n* = 88)	50 (57%) *	38 (44%)	64 (74%)	25 (29%)	8 (9%)	8 (9%)	7 (8%)	10 (11%)	21 (24%)	26 (30%)	28 (32%)	21 (24%)	22 (26%)
No diabetes + normal GMA (*n* = 384)	171 (45%)	159 (41%)	247 (64%)	132 (34%)	81 (21%) **	61 (16%)	23 (6%)	65 (17%)	108 (28%)	117 (30%)	109 (28%)	101 (26%)	115 (30%)
*p* value	0.044	0.72	0.078	0.37	0.0097	0.064	0.1	0.13	0.5	>0.99	0.52	0.79	0.51
Diabetes + hypernormal (*n* = 36)	21 (58%)	16 (44%)	24 (67%)	15 (42%)	7 (19%)	8 (22%)	3 (8%)	6 (17%)	9 (25%)	12 (33%)	12 (33%)	10 (28%)	10 (28%)
No diabetes + hypernomal (*n* = 135)	70 (52%)	70 (52%)	90 (67%)	47 (35%)	31 (23%)	26 (19%)	7 (5%)	26 (19%)	46 (34%)	46 (34%)	42 (31%)	35 (26%)	38 (28%)
*p* value	0.57	0.45	>0.99	0.44	0.822	0.64	0.44	0.81	0.32	0.32	0.84	0.83	>0.99
Diabetes + normal GMA dys-rhythmia (*n* = 85)	39 (46%)	23 (27%)	62 (73%)	20 (24%)	9 (11%)	15 (18%)	3 (4%)	7 (8%)	19 (22%)	23 (27%)	28(33%)	15 (18%)	17 (20%)
No Diabetes + normal GMA dys-rhythmia (*n* = 344)	142 (41%)	134 (39%)	229 (67%)	107 (31%)	65 (19%)	59 (17%)	20 (6%)	71 (21%) **	97 (28%)	102 (30%)	98 (28%)	74 (22%)	102 (30%)
*p* value	0.46	0.27	0.33	0.18	0.078	0.87	0.59	0.0072	0.33	0.69	0.42	0.55	0.07

* *p* < 0.05, ** *p* < 0.01 calculated using Chi-square test. PPD—postprandial fullness, LOA—Loss of appetite, GERD: Gastroesophageal reflux disease, CNV: Nausea and vomiting.

## Data Availability

All data generated or analyzed during this study are included in this article. The additional raw data files can be obtained from the corresponding author upon request through email.

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
