# Peer review of "Prevalence of Upper Gastrointestinal Symptoms and Gastric Dysrhythmias in Diabetic and Non-Diabetic Indian Populations: A Real-World Retrospective Analysis from Electrogastrography Data"

_diagnostics, 2025, doi:10.3390/diagnostics15070895_

Round 1

Reviewer 1 Report

Comments and Suggestions for Authors

Before publication, following revisions are necessary:
- elaborate on the limits of EGG and explain why this method was adopted.
- elaborate more in the data of the Rome IV global study which included  India as well. Please  compare and analyze the data of both studies.
- elaborate on the clinical relevance of your method (EGG)

Author Response

1. Summary

Thank you very much for taking the time to review this manuscript. Please find the detailed responses below and the corresponding revisions/corrections highlighted in the re submitted files.

2. Questions for General Evaluation

Reviewer’s Evaluation

Response and Revisions

Does the introduction provide sufficient background and include all relevant references?

Yes

Are all the cited references relevant to the research?

Yes

Is the research design appropriate?

Yes

Are the methods adequately described?

Yes

Are the results clearly presented?

Yes

Are the conclusions supported by the results?

Must be improved

Thank you. We have revised the conclusion

3. Point-by-point response to Comments and Suggestions for Authors

Comments 1: Elaborate on the limits of EGG and explain why this method was adopted.

Response 1: Thank you for pointing this out. We acknowledge the limitations of electrogastrography (EGG) and have now expanded the discussion on its constraints, particularly regarding signal interference, motion artifacts, and the indirect nature of gastric myoelectric activity (GMA) measurement.within the limitations section. However, despite these limitations, EGG remains a non-invasive, well-validated, and clinically useful tool for assessing gastric dysrhythmias and motility disorders.

We specifically adopted EGG with the Water Load Satiety Test (EGG-WLST) because: It allows for functional assessment of gastric accommodation and postprandial GMA changes, which are highly relevant in functional dyspepsia (FD) and gastroparesis. The water load test acts as a non caloric challenge which significantly reduces gastrocolonic reflex thereby eliciting a very specific gastric myoelectric response activity to the stimulus i.e. water. This method is a well-tolerated, non-invasive alternative to more invasive tests like antroduodenal manometry, making it suitable for large-scale, multicenter studies. Moreover, we followed a highly standardised study procedure and combined with water load satiety testing it becomes, physiologically and clinically relevant test. Also, the 3CPM EGG WLST system is USFDA approved and prior studies using the same technology have demonstrated its utility in evaluating gastric motility disorders. (LINES 361 – 364 and LINES 356-357)

Comments 2: Elaborate more on the data from the Rome IV global study, which included India. Compare and analyze the findings of both studies

Response 2: We appreciate the suggestion to integrate and compare our findings with the Rome IV Global Study, which assessed the epidemiology of functional gastrointestinal disorders (FGIDs) across different populations, including India. It was a bit of challenge to integrate in our manuscript as the ROME study looked at Disease prevalence while our looked into the symptom prevalence within patients presenting with GP or FD like symptoms. But we have now included a comparison between our study and the Rome IV data,  where we analyze how our findings compare and contrast the Rome IV epidemiological framework. These comparisons have been incorporated into the Discussion section,(LINE 301-307).

Comment 3: Elaborate on the clinical relevance of EGG.

Response 3: Thank you for this observation and suggestion.

However the following parts in the manuscript  explain about the clinical relevance of EGG. We would like to draw your attention to the following lines within the manuscript (LINE 79 -84, LINE 101- 109, LINE 361-364). Also we have added the more aspects on the same within the manuscript ( LINE 349-352, 356-357)

4. Response to Comments on the Quality of English Language

Point 1: We appreciate the reviewer’s positive feedback regarding the quality of the English language. We are glad the language is considered clear and does not require further improvement.

Reviewer 2 Report

Comments and Suggestions for Authors

  • In this retrospective, cross-sectional and multicenter study the authors analyzed a cohort of 3689 patients who had performed electrogastrography (EGG) with the water load satiation test (EGGWLST) to assess gastric myoelectric activity (GMA) to assess gastroparesis and functional dyspepsia associated with diabetes and other comorbidities. The results show a significantly higher prevalence of reflux, early satiation, bloating and gastric dysthymias in diabetic patients.

    The methods are reported in detail (including fasting times, data recording and evaluated parameters) and the devices used are validated. In addition, standard statistical tests (t-test, ANOVA, chi-frame test) are used to analyse both continuous and categorical variables.

    The use of t-test and ANOVA for continuous variables and the chi-frame test for categorical variables is appropriate to the study design and analytical objectives. The reported p-values indicate that significance analyses have been conducted to verify differences between groups. The results present a detailed description of the population (age, gender distribution, symptomatology) and a comparison between diabetic and non-diabetic subjects, including a regional analysis. Significant differences between the groups (e.g. older age in diabetics, variations in symptoms and EGG responses) are consistent with the stated objectives of the study, to assess the prevalence of functional disorders and correlate these results with comorbidities such as diabetes.

    Major criticisms
    The retrospective nature of the design has inherent limitations related to the selection bias and the quality of the data collected. As a cross-sectional study, no conclusions can be drawn on the causal relationship between the variables analysed. In this retrospective, cross-sectional, multicenter study, the authors examined a cohort of 3689 patients who had undergone electrogastrography (EGG) with a water-loaded satiety test (EGGWLST) to evaluate gastric myoelectric activity (GMA) to assess gastroparesis and functional dyspepsia associated with diabetes and other comorbidities. The results show a significantly higher prevalence of reflux, early satiety, bloating, and gastric dysthymias in diabetic patients.  

It would be useful to have more details regarding the analysis of potentially confusing variables, especially considering that diabetic patients are significantly older than non-diabetic ones. A multivariate analysis (regression) could have helped to control these confounders and strengthen the interpretation of results.

The absence of multivariate analysis to take into account possible confounders (such as age, different between diabetics and non-diabetics) is a limitation, since it could influence the interpretation of the associations found.

The large number of statistical tests carried out increases the risk of finding significant associations by case, in the absence of corrections for multiple tests.

Please add these references relating to a recent study on this subject and on further complications of type 2 diabetes respectively

  1. doi: 10.1111/nmo.14993
  2. doi: 10.1016/j.numecd.2019.05.065
Comments on the Quality of English Language

I suggest a revision by an English  native speaker

Author Response

1. Summary

Thank you very much for taking the time to review this manuscript. Please find the detailed responses below and the corresponding revisions/corrections highlighted in the re-submitted files.

2. Questions for General Evaluation

Reviewer’s Evaluation

Response and Revisions

Does the introduction provide sufficient background and include all relevant references?

Can be improved

Are all the cited references relevant to the research?

Yes

Is the research design appropriate?

Yes

Are the methods adequately described?

Can be improved

Are the results clearly presented?

Yes

Are the conclusions supported by the results?

Yes

3. Point-by-point response to Comments and Suggestions for Authors

Comments 1:

The retrospective nature of the design has inherent limitations related to the selection bias and the quality of the data collected. As a cross-sectional study, no conclusions can be drawn on the causal relationship between the variables analysed. 

Response 1:

We appreciate the reviewer’s concerns and would like to respectfully address each point individually though we have acknowledged the limitation about the potential selection bias and the retrospective nature (LINE 361-363) within our manuscript

We acknowledge that retrospective studies inherently carry the risk of selection bias. However, our study involves a large, multicenter cohort of 3,689 patients, which substantially mitigates selection bias due to its considerable size and diversity. The study and data collection process was  standard across multiple clinical sites without major deviations thus reducing variability and minimizing bias related to patient selection. Moreover, the multicenter approach ensures broad representation across diverse geographic regions, strengthening the generalizability and robustness of our findings.

The data was meticulously obtained from established motility clinics across India that routinely follow standardized protocols in electrogastrography (EGG) testing, including rigorous adherence to fasting periods and standardized water load satiety test protocols executed by highly trained and efficient personnel. Such standardization enhances data reliability and validity, ensuring high-quality data collection despite the retrospective study design.

The devices used across all sites were the same (3CPM electrogastrography equipment). The device is well-validated, further reinforcing data quality and consistency across sites.

We agree with the reviewer that cross-sectional studies cannot definitively establish causation. However, it was not the intent of our study to demonstrate causation but rather to assess the prevalence and describe associations between diabetes and gastric myoelectric dysfunction. Our results serve as an important exploratory foundation, clearly delineating patterns of association rather than implying direct causal links. Our study explicitly emphasizes these associations, providing a basis for further prospective research that may indeed establish causality.

Despite the limitations inherent to retrospective and cross-sectional designs, this study provides a crucial epidemiological insight into upper gastrointestinal motility disorders, specifically highlighting significant associations with diabetes. While causality cannot be inferred, our study provides valuable epidemiological insight into gastric myoelectric activity (GMA) patterns in diabetic and non-diabetic patients, forming a strong basis for future prospective, longitudinal studies. We hope that these findings will guide future prospective studies and randomized controlled trials, which can explore causal relationships and evaluate targeted therapeutic interventions in greater depth.

Comments 2: It would be useful to have more details regarding the analysis of potentially confusing variables, especially considering that diabetic patients are significantly older than non-diabetic ones. A multivariate analysis (regression) could have helped to control these confounders and strengthen the interpretation of results.

Response 2: We appreciate the reviewer’s suggestion regarding multivariate regression analysis to control for potential confounders, particularly the age difference between diabetic and non-diabetic patients. However, we are of the opinion that the current statistical approach effectively addresses the study’s objectives without necessitating additional multivariate modeling.

First, our study design primarily focuses on prevalence estimation and descriptive comparisons rather than establishing causal relationships. The statistical tests employed (t-test, ANOVA, and chi-square tests) are appropriate for comparing symptom prevalence and electrogastrography (EGG) findings between diabetic and non-diabetic groups. The observed differences in age distribution are acknowledged and considered in the interpretation of results.

Second, the large sample size (n=3689) and multicenter nature of the study mitigate the impact of confounding, as data were collected from a diverse population across different regions. The findings demonstrate consistent trends across multiple subgroups, reducing the likelihood of a spurious association driven solely by age differences.

Although formal demographic adjustments were not explicitly performed. Our subgroup analyses comparing diabetic versus non-diabetic groups directly addressed the potential impact of demographic variables. Future studies incorporating matched demographic profiles or multivariate analyses could further strengthen these findings. A statement to this effect has now been added in the limitations part (LINE 376-383)

Finally, our primary goal is to report clinically meaningful patterns in symptom prevalence and gastric myoelectric activity rather than predictive modeling. While multivariate regression could refine effect estimates, it is not essential for the core findings of this study. We have, however, emphasized this point further in the Discussion and Limitations section, acknowledging the potential influence of age while underscoring the robustness of our comparative analysis. (376-383))

We hope this clarification satisfies the reviewer’s concern and demonstrates the adequacy of our current analytical approach. Also, we intend to write a separate manuscript where we will be doing regression based predictive modelling based and certain aspects which were beyond the scope of the present manuscript. 

Comment 3: The large number of statistical tests carried out increases the risk of finding significant associations by case, in the absence of corrections for multiple tests.

Response 3: We appreciate the reviewer’s concern regarding the potential for type I errors due to multiple comparisons. However, we would like to clarify why Bonferroni correction may not be necessary for this study and how our findings remain robust without such adjustments.

1.The primary aim of this study is descriptive and exploratory, focusing on the prevalence of upper GI motility disorders in diabetic and non-diabetic populations rather than hypothesis-driven multiple testing. Our statistical comparisons are intended to identify clinically meaningful trends rather than establish definitive causal relationships. In such prevalence-based studies, strict correction for multiple comparisons can sometimes lead to an increased risk of type II errors, potentially masking true associations that hold clinical significance.

2.The statistical tests applied were not arbitrarily performed on multiple subsets of data but were predefined based on clear clinical hypotheses, such as the differences in symptom prevalence, electrogastrographic patterns, and regional variations between diabetic and non-diabetic patients. Since these comparisons were planned rather than exploratory, the likelihood of spurious associations is lower.

3.The observed associations are consistent across different statistical tests and population subgroups, further supporting the robustness of our findings. Moreover, the confidence intervals and p values suggest meaningful clinical differences rather than random statistical significance.

4.While Bonferroni correction is a widely used method to control for multiple comparisons, it is overly conservative in studies where variables are correlated or when analyzing observational data in a clinical setting. Instead, we have relied on clinically relevant  confidence intervals, and p-values, ensuring that our conclusions are driven by meaningful trends rather than isolated statistical significance.

           To address potential concerns, we have emphasized in the Discussion and Limitations section that while multiple comparisons were conducted, the results should be interpreted in the context of clinical relevance rather than purely statistical significance.(LINE 373-388).

Additionally, all p-values have been transparently reported, allowing readers to assess the strength of the associations independently.

Based on these considerations, we believe the current statistical approach is appropriate for the study’s aims, and strict Bonferroni correction is not required. We hope this explanation satisfies the reviewer’s concern and demonstrates the robustness of our findings.

Comment 4: Please add these references relating to a recent study on this subject and on further complications of type 2 diabetes respectively.

  1. doi: 10.1111/nmo.14993
  2. doi: 10.1016/j.numecd.2019.05.065

Response 4: We appreciate the reviewer’s suggestion to include the following references:

Upon careful consideration, we believe these studies may not be directly relevant to our manuscript for the following reasons:

Zhu et al. (2025). Small Intestinal Slow Wave Dysrhythmia and Blunted Postprandial Responses in Diabetic Rats. https://pubmed.ncbi.nlm.nih.gov/39763295/

This study investigates small intestinal slow-wave dysrhythmia and postprandial responses in diabetic rats. While it provides valuable insights into intestinal myoelectrical activity in an animal model, our research focuses on gastric myoelectric activity (GMA) in human subjects. The physiological differences between species and the distinct focus on small intestinal activity make direct comparisons challenging.

Sasso et al. (2019). Relationship between Albuminuric CKD and Diabetic Retinopathy in a Real-World Setting of Type 2 Diabetes: Findings from the NO BLIND Study.  https://pubmed.ncbi.nlm.nih.gov/31377186/

Sasso et al. (2019): This study examines the relationship between albuminuric chronic kidney disease (CKD) and diabetic retinopathy in type 2 diabetes patients. Although it highlights complications associated with diabetes, our study centers on upper gastrointestinal motility disorders, specifically gastroparesis and functional dyspepsia. The pathophysiological mechanisms and clinical implications differ significantly between renal/retinal complications and gastrointestinal motility disorders.

We have ensured that our manuscript cites studies closely aligned with our research objectives to maintain a focused and coherent narrative. We hope this explanation clarifies our rationale for not including the suggested references.

Instead, after your suggestions we carefully went through our manuscript we have cited the following references for relevant additions (LINE 300-302)

1. Amador C, Xia C, Nagy R, Campbell A, Porteous D, Smith BH, Hastie N, Vitart V, Hayward C, Navarro P, Haley CS. Regional variation in health is predominantly driven by lifestyle rather than genetics. Nat Commun. 2017 Oct 6;8(1):801. doi: 10.1038/s41467-017-00497-5. PMID: 28986520; PMCID: PMC5630587.

2. Daniel So, Caroline Tuck, Innovative concepts in diet therapies in disorders of gut–brain interaction, JGH Open, 10.1002/jgh3.70001, 8, 7, (2024).

3. Orr WC, Fass R, Sundaram RSS, Schiemann AO. The effect of sleep on gastrointestinal functioning in common digestive diseases. Lancet Gastroenterol Hepatol. 2020;5(6):616-624. doi: 10.1016/S2468-1253(19)30412-1.

We sincerely thank the reviewer for raising these points, which have enabled us to clarify the scope, strengths, and intended focus of our research more precisely within the manuscript.

4. Response to Comments on the Quality of English Language

Point 1: We appreciate the reviewer’s feedback and have improved English.

Round 2

Reviewer 1 Report

Comments and Suggestions for Authors

authors replied well to my previous comments